# DT-RRNS: Routing Protocol Design for Secure and Reliable Distributed Smart Sensors Communication Systems [note 1]

**DOI:** 10.3390/s23073738

**Published:** 2023-04-04

**Authors:** Andrei Gladkov, Egor Shiriaev, Andrei Tchernykh, Maxim Deryabin, Mikhail Babenko, Sergio Nesmachnow

**Affiliations:** 1Faculty of Mathematics and Computer Science, North-Caucasus Federal University, 355017 Stavropol, Russia; 2Computer Science Department, CICESE Research Center, Ensenada 22860, Mexico; 3Control/Management and Applied Mathematics, Ivannikov Institute for System Programming, 109004 Moscow, Russia; 4Computing Platform Lab, Samsung Advanced Institute of Technology, Suwon 16678, Republic of Korea; 5North-Caucasus Center for Mathematical Research, North-Caucasus Federal University, 355017 Stavropol, Russia; 6Faculty of Engineering, Universidad de la República, Montevideo 11300, Uruguay

**Keywords:** smart city, Residue Number System, Secret Sharing Schemes, distributed transmitted, reliability, Mobile Ad hoc Network, communication, heterogeneous sensor networks

## Abstract

A smart city has a complex hierarchical communication system with various components. It must meet the requirements of fast connection, reliability, and security without data compromise. Internet of Things technology is widely used to provide connectivity and control solutions for smart sensors and other devices using heterogeneous networking technologies. In this paper, we propose a routing solution for Wireless Sensor Networks (WSN) and Mobile Ad hoc NETworks (MANET) with increasing speed, reliability, and sufficient security. Many routing protocols have been proposed for WSNs and MANETs. We combine the Secret Sharing Schemes (SSS) and Redundant Residual Number Systems (RRNS) to provide an efficient mechanism for a Distributed dynamic heterogeneous network Transmission (DT) with new security and reliability routing protocol (DT-RRNS). We analyze the concept of data transmission based on RRNS that divides data into smaller encoded shares and transmits them in parallel, protecting them from attacks on routes by adaptive multipath secured transmission and providing self-correcting properties that improve the reliability and fault tolerance of the entire system.

## 1. Introduction

Internet of Things (IoT) technology for a smart city is widely used to provide solutions for connecting smart things using heterogeneous networks and advanced communication technologies.

As the main assets, various city systems, objects, and sensors can act as distributed information systems generating data, i.e., power plants, schools, transport, law enforcement agencies, hospitals, and other public services. The main objective is to improve the living standard and urban service quality. The information gained from smart sensors allows us to analyze and manage the urban environment in real-time with a quick response. There are many scientific, commercial, and governmental solutions for implementing a smart city concept.

According to Deakin’s generalized definition [1], a smart city is a city that uses an information system to meet the needs of city residents. It is not only a set of technological solutions but is the application of these technologies by local communities.

Let us consider the main hardware components of the smart city network. It consists of many elements, including video surveillance, emergency call systems, biometric systems, city and banking services, intelligent transport, and IoT solutions (Radio Frequency Identification [2,3], sensors for measuring temperature, humidity, illumination, pressure, etc.). Smart sensor networks play a substantial role in IoT. Their components include sensing, data collection, heterogeneous connectivity, data processing, etc. [3,4,5].

Large-scale data sharing in a distributed environment is fraught with data security and privacy issues, as data being compromised can harm people and the entire system. Another important aspect is reliability [3]. Failures can delay the response of emergency systems, medical, and rescue services. Thus, when building a smart city communication infrastructure, design methods that provide data security at the required level while having high reliability and speed are very important.

In [6], we propose combining the Secret Sharing Schemes (SSS) and Redundant Residual Number Systems (RRNS) as an efficient security mechanism for a smart city dynamic heterogeneous network and show how RRNS increases communication reliability through effective correcting management.

This paper presents a more extensive and in-depth study of data transmission in the proposed DT-RRNS protocol. We propose a routing solution for the Wireless Sensor Network (WSN) and Mobile Ad hoc NETwork (MANET) and present a methodology for ensuring the security and reliability of data transmission.

The method is based on Node-Disjoint Multipath Routing [5], which allows to exchange and manage data between smart things, ensuring privacy by the threshold SSSs and Redundant Residue Number System (RRNS). We describe details of the generation of parameters that overcome the limitations of the well-known Mignotte scheme, data partitioning, and data recovery and provide a theoretical analysis of reliability and security bounds.

We consider the network as a distributed infrastructure rather than a centralized system. It is well known that for large networks, centralized data processing imposes a large load on the central computing bottleneck slowing down the entire system. More detailed arguments about the positive and negative properties of a decentralized network can be seen in [7,8,9].

SSS is a cryptographic technique that splits a secret into several shares $s=\left\{s_{1},s_{2},\ldots ,s_{n}\right\}$ and distributes them among participants. In the most used (k,n) threshold SSS, a combination of k shares from n is needed to recover the secret, where k≤n.

RRNS is one of the most common non-positional number systems that represents the number of a positional system as a tuple of n numbers $\left(s_{1},s_{2},\ldots ,s_{n}\right)$, obtained by dividing numbers into residuals (see Section 4). Among many of its applications, we could mention the acceleration of operations due to the parallel implementation of basic arithmetic, information integrity control, digital signal processing, etc.

This paper is structured as follows: Section 2 considers data transmission in smart city IoT networks. Section 3 discusses existing approaches to ensure security, as well as the advantages of distributed SSS schemes based on RRNS. Section 4 describes the RRNS and SSS details. Section 5 discusses the proposed DT-RRNS. Section 6 presents the proof of correctness and discusses the main properties of the proposed scheme, its security, and its reliability. Section 7 discusses a generalized scheme and principles for secure and reliable data transmission. Section 8 analyzes data transmission security. Section 9 provides a performance analysis. Section 10 presents the main conclusions and future work.

## 2. Data Transmission in IoT Networks

A wireless ad hoc network and MANET are important concepts of smart city communication. It is widely used for ensuring self-configuring and dynamic connectivity between sensors, humans, and devices that send and receive information.

Lobo et al. [10] study the Quality of Service of MANET in smart city networks with an emphasis on healthcare. Several frameworks were considered that improve the transmission quality of MANET, as well as individual elements, such as video signal transmission. Cardone et al. [11] discuss the MANET and WSN hybrid network for fast data collection in the smart city. The authors provide a transmission protocol based on modern data transmission standards considering IPv6. Pandey et al. [12] study methods to improve the reliability of MANET networks and propose a method of self-healing nodes.

In this work, our goal is to increase the speed and reliability of MANET communication to ensure security. To achieve this goal, we propose the use of RRNS in MANET.

In the original version, MANET solves the minimax optimization problem of finding the shortest path in the network. The smart city network can be represented as a directed graph, where the vertices are the communication nodes (devices in the network), and the arcs are the data transmission between the nodes. Let us establish that G(V,E) is a network graph with a flow $v_{0}\in V$ and path cost function c:E→R. We assume that the set of vertices V split into two non-overlapping subsets $V_{A}$ and $V_{B}$ $\left(V_{A}\cup V_{B}=V,V_{A}\cap V_{B}=\emptyset \right)$.

Now, we fix a pair of mappings:(1)$$s_{A}:v\to V_{G}\left(v\right)\;for\;v\in V_{A}\backslash \left\{v_{0}\right\};s_{B}:v\to V_{G}\left(v\right)\;for\;v\in V_{B}\backslash \left\{v_{0}\right\};$$
where $V_{G}\left(v\right)$ is the set of ends of all arcs outgoing from a vertex v. We define the following subgraph $T_{s}=\left(V,E_{s}\right)$, generated by a set of arcs of the form $\left(v,s_{A}\left(v\right)\right)$ and $\left(v,s_{B}\left(V\right)\right)$. This subgraph has the property that for some given vertex w∈V, or there is a way $P_{T},\left(w,v_{0}\right)$ from w to $v_{0}$.

For an arbitrary vertex, w∈V defines the value $\tilde{c}\left(s_{A},s_{B},w\right)$ as the sum of the costs of the arcs of the path $P_{T},\left(w,v_{0}\right)$, if such a path exists in $T_{s}$. If the path $P_{T},\left(w,v_{0}\right)$ does not exist in $T_{s}$, we assume the value $\tilde{c}\left(s_{A},s_{B},w\right)$ equals to ∞ or −∞ depending on the positivity and negativity of the sum of the costs of the arcs of the oriented cycle $C_{w}$.

If the sum of the costs of the arcs of an oriented cycle $C_{w}$ is zero, then $\tilde{c}\left(s_{A},s_{B},w\right)$ equals the sum of the costs of the arcs of the path connecting the vertex w with the cycle $C_{w}$. That is, a problem is formulated as $F\left(w\right)=\underset{s_{A}}{\min}\underset{s_{B}}{\max}\tilde{c}\left(s_{A},s_{B},w\right)$.

Let us consider the data transmission model presented in Figure 1 and Figure 2. It is known that MANET transmits data using devices located on the infrastructure-less, distributed wireless networks without static-located transmission stations. It is an interesting and promising solution providing communication of a big variety of devices, from mobile devices to personal cars, from smart devices to public transport, etc. In addition, a smart city infrastructure also contains static nodes, such as data centers, storage, decision centers, etc.

For such a dynamic heterogeneous network, we propose the concept of parallel data transmission based on RRNS that divides data into smaller shares and transmits them in parallel. The self-correcting properties of RRNS can improve the reliability and fault tolerance of the entire system [13,14,15,16].

Figure 1 shows a conceptual model described above. We group the elements of a smart city according to common features. They can be separated from each other by large distances and distributed like data management modules.

This model gives a general idea of the transmission network complexity. Each group of components is connected to other groups, and control devices can communicate with any device on the network. In such a data transmission model, MANET provides a definite advantage. Devices, such as sensors, can send data to a destination, transmitting it through other devices within the network.

Figure 2 shows the data transmission from the sensor to the recipient in the DT model. The recipient can be a data warehouse, decision center, data processing center, cloud data analysis, etc. RRNS transmits data in the MANET network in parallel breaking the message into several shares. It improves the speed at which data are transmitted across communication channels since such shares are smaller than the original message.

We use the term Weighted Number System (WNS) as a traditional positional decimal number system.

Figure 3 shows the model of data transmission packets. The receiver collects shares of information and combines them. The application knows how many shares have arrived and how many shares should arrive. RRNS has self-correcting properties for recovering the message if one or several shares are lost or intentionally changed. If arrived shares are not enough, it waits a certain time, and the packet is requested again or ignored. As a result, we can have a network with increasing speed and reliability.

## 3. DT Security and Reliability

Our main approach is to use RRNS to ensure the security of data transmission. Let us discuss and compare well-known solutions for providing reliability and security of distributed data storage and transmission. Four main methods are used to ensure reliability [17]: Replication, Erasure code, Erasure code modifications, and Error correction code.

Chang et al. [18] presented a modified data replication method, providing a high encoding and decoding speed. However, it requires additional cryptographic primitives to ensure security and has a high redundancy compared to erasure codes.

Many different modifications of erasure codes have been proposed to create reliable methods for DT. The joint use of error correction and erasure codes maintains system performance and minimizes the load on the data transmission network when recovering lost fragments [19,20].

Erasure codes based on the RRNS [21] allow data to be processed in the encoded form [14]. So, it can be used both in the design of low-power wireless data transmission devices and DT.

Secure DTs are based on the use of cryptographic primitives—symmetric encryption algorithms (AES) and digital signatures based on RSA (Rivest, Shamir, Adleman) [22]. The advantages of these approaches are high speed of encryption and decryption and low data redundancy. The disadvantage is that an error in the encrypted data leads to its loss. To eliminate this shortcoming, the use of additional mechanisms for accessing data for a long time is required [23].

When building secure and reliable DT, the following methods are used: elliptic cryptography and erasure codes [24,25], access structures [26,27], error correction codes [28,29], graph-based algorithms and modified data replication algorithm [30], attribute-based encryption [31], etc.

An alternative approach is to use recovery codes [20], erasure codes, and error correction codes based on RRNS [19]. However, recovery codes and erasure codes do not allow encoded data processing. Homomorphic calculations process encoded data without additional computational costs for decoding.

A significant breakthrough in the field of homomorphic computing came from the work of Gentry [32]. The authors proposed a fully homomorphic scheme to perform both addition and multiplication. The main disadvantages of this algorithm are significant data redundancy and lack of control over the results of arithmetic operations.

Particular attention should be paid to the distributed data storage model proposed in [33], guaranteeing security, privacy, homomorphism, reliability, and scalability. The authors propose two approaches to building systems based on homomorphic access structures in RRNS, with RRNS moduli being used as secret keys stored by users. Data processing leads to an exponential increase in the load on the network and memory, which makes this model inapplicable in practice in modern conditions.

Access structures [34,35] ensure data security and confidentiality. RRNS implements the same functionality as the Mignotte scheme but allows you to control the results of data processing. DT is also characterized by collusion risks [36]. Several approaches have been developed to prevent cloud collusion [26]. As mentioned above, the non-stationarity of the cloud environment reduces the efficiency, performance, reliability, and security of the system. The adaptive paradigm reduces uncertainty but is rarely used in cloud computing [36].

Let us consider the following scenario. The user has confidential data and decides not to send it using a single path. He divides them into several shares and transmits them in different paths between nodes. There are several types of security threats in this scenario.

Deliberate threats include unauthorized access to information, interception, falsification, hacker attacks, etc., in one or more nodes.

Random threats include errors, crashes, etc. They can lead to the loss of one or more shares of data, inconsistencies between different copies of the same data, and/or the inability to recover the original data. Collusion threats are illegal agreements between two or more adversaries (in the context of different paths between nodes, the adversaries are nodes) to gain full access to personal data. Cryptographic protocols can be used to mitigate the risks of deliberate threats, but this is not enough for random threats.

We consider reliability and security as close concepts of an information violation. Therefore, statements related to reliability are used to discuss security and vice versa.

To improve the security and reliability of data transmission systems, DT is based on access structures and error correction codes. It transmits data through various paths between nodes and minimizes the chance of information theft or loss in case of intentional and accidental threats.

In the next sections, we show how the size of shares and their number can change the reliability, security, speed, etc. of data transmission. These structures reduce the load on the transmission network compared to the classical replication mechanism and reduce the cost.

## 4. Residue Number System and Secret Sharing

(k, n)-RRNS is determined by a system of pairwise coprime moduli $\left\{p_{1},p_{2},\ldots ,p_{n}\right\}$. Positional integer number s such that 0≤s<P, where $P=\overset{k}{\underset{i=1}{\prod }}p_{i}$, is represented as a tuple of n numbers $s\overset{RRNS}{\to }\left(s_{1},s_{2},\ldots ,s_{n}\right)$, where n=k+r and
(2)$$s_{i}=s\mathrm{mod}p_{i},\;i=1,2,\ldots ,n.$$

RRNS is a redundant representation of the Residue Number System (RNS). Redundancy is represented by additional moduli in the moduli set. k is the RNS dimension; r is the dimension of redundant moduli; and n is the RRNS dimension. According to the RRNS property, if the number of moduli is r, then it can detect r and correct ⌊r/2⌋ errors.

Redundancy supports reliable data processing and transmission systems with multiple error detection and correction. To detect and correct errors in RRNS, several methods are used, for instance, syndrome and projection methods [28,29]. If we consider RRNS not only as the error detection, localization, and correction code but also as the Mignotte SSS, then we can conclude that RRNS ensures data security.

RRNS has many applications because of its properties such as parallelism and modularity, among which we can mention hardware and software acceleration, information integrity control, digital signal processing, increasing the robustness of information transmission between computers, etc.

Modular calculus is based on the Chinese Remainder Theorem (CRT) [28], according to which the number s can be uniquely calculated by the formula
(3)$$s={\left|\overset{k}{\underset{i=1}{\sum }}{\left|P_{i}{}^{-1}\right|}_{p_{i}}P_{i}s_{i}\right|}_{P},$$
where $P_{i}=\frac{P}{p_{i}}$, ${\left|P_{i}{}^{-1}\right|}_{p_{i}}$—multiplicative inversion $P_{i}$ modulo $p_{i}$, for i=1,2,…,k.

This method is called the CRT method or the Garner method. However, it is computationally complex, since it requires division by a sufficiently large number P. It is worth noting that there are many well-developed methods for an efficient implementation of calculating the remainder of the division and converting numbers back from RRNS to a WNS. It makes this system suitable for use as the basics of a SSS [15,26,28].

Let us consider SSSs using Shamir’s threshold scheme as an example [27]. The idea of this scheme is that the secret is represented as a polynomial k−1 degrees. Then, to interpolate the resulting polynomial, it is necessary k points, and the polynomial can be divided into n shares. Then, the secret-sharing process is as follows. Let we need to divide the secret s on *n* shares. To do this, take a prime number p>s. The following polynomial is constructed:(4)$$F\left(x\right)=\left(a_{k-1}x^{k-1}+a_{k-2}x^{k-2}+\ldots +a_{1}x+s\right)\;\mathrm{mod}\;p,$$
where $a_{k-1},a_{k-2},\ldots ,a_{1}$—random numbers that are only known when the secret is shared.

The secret recovery occurs due to the calculation of the Lagrange interpolation polynomial according to the following formula:(5)$$F\left(x\right)=\sum_{i}l_{i}\left(x\right)y_{i}\;\mathrm{mod}\;p;l_{i}\left(x\right)=\prod_{i\neq j}\frac{x-x_{j}}{x_{i}-x_{j}}\mathrm{mod}\;p,$$
where $\left(x_{i},y_{i}\right)$—polynomial point coordinates. In addition, there is a limitation: all calculations are performed only in the final field p. In this scheme, an integer polynomial is used. Despite the low redundancy and high scalability, the field space p is not used efficiently.

This scheme was developed by Hugo Krawczyk in 1993 [37]. In this scheme, integer coefficients are shares. It is a (k,n) threshold SSS. It distributes s among k participants randomly. The recovery of the secret is possible from k shares, while k−1 shares do not allow you to recover s.

Let us consider the Information Dissemination Algorithm designated as IDA (Algorithm 1). This algorithm works for parameters n (total number of shares) and k (required number of shares for recovery). It includes a secure encryption function with a private key, which is designated as ENC. In addition, the algorithm implements a computationally secure (k, n) SSS. It is also worth noting that the space of both the secret and the message in this scheme is the same as for the encryption function ENC.
**Algorithm 1**. Secret Sharing of Krawczyk scheme.1. Choosing a random encryption key K; secret s is encrypted by ENC, $e={\mathrm{ENC}}_{K}\left(s\right)$.2. e is divided into n fragments—$e_{1},\;e_{2},...,\;e_{n}$ by the scheme.3. K is represented as a tuple of n numbers $K_{1},\;K_{2},\ldots ,\;K_{n}$ by Asmuth-Bloom SSS. 4. Shares $m_{i}=\left(e_{i},K_{i}\right)$, $i=\bar{1,n}$ are distributed between participants

In Algorithm 2, every share $m_{i}$ has a bit length $\left|e_{i}\right|+\left|K_{i}\right|$, where |x| is a bit number of x. Evidence of this, as well as confirmation of the secrecy of the scheme, is given in [38].
**Algorithm 2.** Secret Recovery of Krawczyk scheme.1. k participants combine their shares $m_{i_{j}}=\left(e_{i_{j}},\;K_{i_{j}}\right)$ with indexes $\left\{i_{1},i_{2},\ldots ,i_{k}\right\}$ together2. e is recovered from shares $e_{i_{j}}$.3. key K is recovered from $K_{i_{j}}$ by Asmuth-Bloom4. Using K, e is decrypted then s is recovered.

Despite the obvious advantages of these schemes (low redundancy, scalability, flexibility), they have several disadvantages, such as the inability to add new participants without recovering the secret and re-sharing it which is important for smart city infrastructure. The advantages and limitations of the DT-RRNS scheme are discussed in the next sections.

## 5. Secret Sharing Scheme with Residue Number System

In this section, we introduce the basic concepts of security of two well-known SSS based on RRNS: Asmuth-Bloom and Mignotte.

Let each participant have a unique number or identifier. The entire set of these numbers we call the universal set of numbers and denote U (in the simplest case U={1,2,…,n}, where n is the number of participants in the scheme).

The set of authorized (qualified) coalitions is called the authorized subsets of U denoted by I. Participants of the qualified subsets can recover the secret from their shares when they act together to pool their knowledge.

An unauthorized subset is a subset $\tilde{I}$ of participants of any coalition that does not have the right to recover the secret.

In the Asmuth-Bloom scheme, $p_{0}$ is a secret key, and $s\in \left[0,p_{0}\right)$. The moduli $p_{1}<p_{2}<\ldots <p_{k}<p_{k+1}<\ldots <p_{n}$ have to be chosen, so that $\overset{k}{\underset{i=1}{\prod }}p_{i}>p_{0}\overset{k-2}{\underset{i=0}{\prod }}p_{n-i}.$ The last inequality is usually called the Asmuth-Bloom condition. At the stage of sharing the secret, a random number $r_{n}$ is generated such that $s^{\prime }=s+r_{n}p_{0}<\overset{k}{\underset{i=1}{\prod }}p_{i}$.

Secret $s^{\prime }$ is divided so that $s_{i}=s^{\prime }\mathrm{mod}p_{i}$ is a share for participant i, where i=1,2,…,n. Any set of authorized participants with numbers from I can recover the secret; wherein n≥|I|=m≥k.

Using the CRT, $s^{\prime }$ is recovered based on its RRNS representation $\left(s_{i_{1}},s_{i_{2}},\ldots ,s_{i_{m}}\right)$ with moduli $p_{i_{1}},p_{i_{2}},\ldots ,p_{i_{m}}$, where $i_{j}\in I$, for all j=1,2,…,m. s is recovered as the remainder of the division of $s^{\prime }$ on $p_{0}$: $s=s^{\prime }\;\mathrm{mod}\;p_{0}$.

Let us consider an unauthorized coalition of participants with numbers from $\tilde{I}$. Then $\left|\tilde{I}\right|\leq k-1$, let $P=\overset{k}{\underset{i=1}{\prod }}p_{i}$ and $\tilde{P}=\underset{i\in \tilde{I}}{\prod }p_{i}$. In this case, $\tilde{s}=s^{\prime }\mathrm{mod}\;\tilde{P}$. According to the Asmuth-Bloom conditions $P/\tilde{P}>p_{0}$ and $\left(\tilde{P},p_{0}\right)=1$. Thus, as shown in [34], an unauthorized coalition obtained fewer than k shares does not receive any useful information about the secret.

In the Mignotte (k, n) threshold scheme, moduli $p_{1}<p_{2}<\ldots <p_{k}<p_{k+1}<\ldots <p_{n}$ are chosen to satisfy the inequality:(6)$$\alpha =\overset{k-2}{\underset{i=0}{\prod }}p_{n-i}<\overset{k}{\underset{i=1}{\prod }}p_{i}=\beta .\;$$

To achieve security, secret s has to be in the interval (α,β). Any set I of authorized participants can recover the secret, wherein |I|=m≥k. s is recovered by CRT using $\left(s_{i_{1}},s_{i_{2}},\ldots ,s_{i_{m}}\right)$ and moduli $p_{i_{1}},p_{i_{2}},\ldots ,p_{i_{m}}$, where $i_{j}\in I$, for all j=1,2,…,m.

To ensure security, Mignotte sequences with a large value (β−α)/β should be used [35]. This scheme is not computationally secure but has practical applications due to reduced redundancy compared with Asmuth-Bloom.

Let us consider the concept of entropy, which plays an important role in SSS security theory.

We denote the entropy of the secret as $H\left(s\right)=\log_{2}s$. In this case, the entropy is maximum. Knowing the subset of the shares $s^{*}=\left(s_{i_{1}},s_{i_{2}},\ldots ,s_{i_{m}}\right)$, we denote entropy as $H\left(s^{*},I\right)=\min\left(\log_{2}s,\underset{i\in I}{\sum }\log_{2}p_{i}\right)$, where $\forall j:i_{j}\in I$ and |I|=m. If I is the set of authorized participants, then $H\left(s^{*},I\right)=H\left(s\right)=\log_{2}s$. The important characteristic of the SSS is the uncertainty of the secret that is defined by
(7)$$\Delta \left(s,s^{*},I\right)=H\left(s\right)-H\left(s^{*},I\right).$$

SSS is computationally secure if $\Delta \left(s,s^{*},\bar{I}\right)=H\left(s\right)$ for all $\bar{I}$, where $\bar{I}$ is the set of unauthorized participants. For the set of authorized participants, the secret can be recovered correctly; hence, the uncertainty is equal to:(8)$$\Delta \left(s_{i}:i\in \tilde{I}\right)=0.$$

To analyze the security of SSS based on RRNS, an additional concept of the perfect SSS was introduced in [38]. SSS is called perfect if any unauthorized subset participants cannot obtain any information about the secret. Hence, the scheme is perfect if, for all unauthorized subsets of participants with numbers $\tilde{I}$ and for any ε>0, there is s, such that, for $p_{0}<p_{1}<\ldots <p_{n}$, $p_{i}>s$ (i=0,1,…,n), and $\Delta \left(s,s^{*},\bar{I}\right)<\varepsilon .$

The scheme is called ideal if the space of share has the same dimension as the secret space. An ideal SSS is perfect with the smallest possible size of each share.

The question of how exactly it is necessary to choose the parameters of the SSS on the RRNS so that it has the asymptotic idealness property remains open. In [38], the authors show the asymptotic idealness of the Asmuth-Bloom scheme using “sufficiently close” coprime numbers for RRNS moduli. The work [39] considers so-called compact sequences of coprime numbers with an initial value $p_{0}$ when $p_{n}<p_{0}+p_{0}^{\theta }$ for some real number θ∈(0,1). In the following analysis, we assume that the compact sequences of coprime numbers are used as the moduli sets.

Let us now consider the concept of computationally secure SSS. Assume that at some point in time, unauthorized participants collect several shares with numbers $\tilde{I}$. The objective of the unauthorized participants is to recover the secret based on the available data.

Let S be a universal set of all subsets of possible secrets recovered from all available shares. S can be divided into two subsets. First subset $S_{1}$ consists of all possible secrets that cannot be used to obtain the secret. The second subset $S_{2}$ contains all remaining possible secrets. For example, if the Mignotte scheme knows the share of the secret $s_{j}$ for modulo $p_{j}$, 0≤j≤n, then the secret must satisfy the condition: $s\equiv s_{j}\mathrm{mod}p_{j}$. Therefore, in this case, $S_{1}=\left\{s:s\in S\wedge s≢s_{j}\mathrm{mod}p_{j}\right\}$ and $S_{2}=\left\{s:s\in S\wedge s\equiv s_{j}\mathrm{mod}p_{j}\right\}$. Note that if the SSS is perfect, then $S_{1}=\emptyset$ and $S_{2}=S$.

Thus, to obtain the original secret, it is necessary to use all combinations of indexes included in $S_{2}$ and the security of the scheme depends on the cardinality of this set and the computational complexity of the complete permutation.

It is necessary to generate the scheme parameters in such a way that unauthorized participants cannot, using modern computing resources, obtain the secret in a reasonable time. A scheme that meets these conditions is called a computationally secure scheme. As a measure of computationally secure, we take the cardinality of the set $S_{2}$: $f\left(\tilde{I}\right)=\left|S_{2}\right|$.

For the Asmuth-Bloom scheme, considering its asymptotic idealness, and Asmuth-Bloom condition, $f\left(\tilde{I}\right)=\left|S\right|\leq p_{0}$ for $\tilde{I}$.

Computationally secure schemes are not always ideal but have reduced redundancy, which is important in practical applications.

## 6. Data Transmission Security and Reliability

Let us consider parameter generation, secret sharing, and secret recovery for threshold (k, n)-DT-RRNS.

Parameter generation. A compact sequence of coprime numbers is selected $p_{1}<p_{2}<\ldots <p_{k}<p_{k+1}<\ldots <p_{n}$; where $p_{n}<p_{1}+p_{1}^{\theta }$ and θ∈(0,1); secret s∈[0,P), where $P=\overset{k}{\underset{i=1}{\prod }}p_{i}$ is the dynamic range of the RRNS.

Secret Sharing. Shares $s_{i}$ of a secret s are calculated as $\forall \;i=\bar{1,n}:s_{i}=s\mathrm{mod}p_{i}$.

Secret Recovery. Any authorized set of participants with numbers I can uniquely recover the secret, where |I|=m≥k. s is calculated using CRT $s={\left|\overset{m}{\underset{j=1}{\sum }}s_{i_{j}}P_{j,m}{\left|P_{j,m}^{-1}\right|}_{p_{i_{j}}}\right|}_{M}$, where $M=\overset{m}{\underset{j=1}{\prod }}p_{i_{j}}$ and $P_{j,m}=M/p_{i_{j}}$.

Let us consider the main properties of the DT-RRNS. The following notations are introduced.
IAuthorized set is subset of {1,2,…,n}, cardinality is equal to k
$\tilde{I}$Unauthorized set is subset {1,2,…,n}, cardinality is less to k
${\tilde{I}}_{max}$Unauthorized set {n−k+2, n−k+3,…,n} cardinality is equal to k−1
$\tilde{P}=\underset{i\in \tilde{I}}{\prod }p_{i}$Dynamic range for an unauthorized set $\tilde{I}$
${\tilde{P}}_{max}=\underset{i\in {\tilde{I}}_{max}}{\prod }p_{i}$Dynamic range for an unauthorized set ${\tilde{I}}_{max}$
$f\left(\tilde{I}\right)$Cardinality of the set of possible secrets $\tilde{s}\;$ for a given $\tilde{P}$
$f\left({\tilde{I}}_{max}\right)$Cardinality of the set of possible secrets $\tilde{s}\;$ for a given ${\tilde{P}}_{max}$
$\tilde{s}=s\;\mathrm{mod}\;\tilde{P}$Projection of the secret s modulo $\tilde{P}$
$f_{k}$Approximate value of $f\left(\tilde{I}\right)$
SUniversal set of all subsets of possible secrets recovered from all available shares$Q_{i}$The probability of intercepting i nodesRProximity of the cardinality of possible secrets $f\left({\tilde{I}}_{max}\;\right)$ to $p_{1}$


The following statement shows a lower bound for the moduli selection.

**Statement** **1.***In DT-RRNS, for any unauthorized subset of participants with numbers* $\tilde{I}$, $P>\tilde{P}=\underset{i\in \tilde{I}}{\prod }p_{i}$ *and* $p_{1}>2^{k-1}$.

**Proof.** Using the fact that in threshold SSS the maximum unauthorized subset is the subset numbered n,n−1,…,n−k+2 and considering the definition of compact sequences, we obtain:(9)$$\tilde{P}<\overset{k-2}{\underset{i=0}{\prod }}p_{n-i}<{\left(p_{1}+p_{1}^{\theta }\right)}^{k-1}=p_{1}^{k-1}{\left(1+p_{1}^{\theta -1}\right)}^{k-1}<p_{1}^{k-1}2^{k-1}.$$On the other hand, $P>p_{1}^{k}$. From here
(10)$$\frac{P}{\tilde{P}}>\frac{p_{1}^{k}}{p_{1}^{k-1}2^{k-1}}=\frac{p_{1}}{2^{k-1}}$$To comply with the condition $P>\tilde{P}$ it is necessary to fulfill the inequality $P/\tilde{P}>1$. This inequality will necessarily hold if the inequality $\frac{p_{1}}{2^{k-1}}>1$, or which is equivalent, $p_{1}>2^{k-1}$. The statement is proven. □

In other words, the DT-RRNS is applicable when choosing a module $p_{1}$ at the parameter generation stage such that $p_{1}>2^{k-1}$.

**Statement** **2.***For DT-RRNS, when combining the shares of an unauthorized subset of participants of* $\tilde{I}$, *the cardinality of the enumeration set *$f\left(\tilde{I}\right)$ *is determined by the expression:*(11)$$\lfloor \frac{P}{\tilde{P}}\rfloor \leq f\left(\tilde{I}\right)\leq \lfloor \frac{P}{\tilde{P}}\rfloor +1,$$*where* $\tilde{P}=\underset{i\in \tilde{I}}{\prod }p_{i}$.

**Proof.** Since the shares, whose numbers belong to $\tilde{I}$, are known, then for all $p_{i_{j}}$ such that $i_{j}\in \tilde{I}$, it is possible to recover the number $\tilde{s}\equiv s\;\mathrm{mod}\tilde{\;P}$ due to $s=a\tilde{P}+\tilde{s}$. The only unknown secret parameter is a∈ℤ.Let us define the upper and lower bound of a.s is defined with dynamic range P. Consequently $0\leq a\tilde{P}+\tilde{s}\leq P-1$, where
(12)$$-\frac{\tilde{s}}{\tilde{P}}\leq a\leq \frac{P-\tilde{s}-1}{\tilde{P}}.\;$$Taking into account that s and a are non-negative and since $\tilde{s}<\tilde{P}$ and $\lfloor \frac{P}{\tilde{P}}\rfloor -1<\frac{P-\tilde{s}-1}{\tilde{P}}<\lfloor \frac{P}{\tilde{P}}\rfloor +1$, $\left\lfloor \frac{P}{\tilde{P}}\rfloor -1\leq \lfloor \frac{P-\tilde{s}-1}{\tilde{P}}\rfloor \leq \lfloor \frac{P}{\tilde{P}}\right\rfloor$. We have
(13)$$0=-\lceil \frac{\tilde{s}}{\tilde{P}}\rceil \leq a\leq \lfloor \frac{P-\tilde{s}-1}{\tilde{P}}\rfloor .$$That is, a lies in between $\left[0,\lfloor \frac{P-\tilde{s}-1}{\tilde{P}}\rfloor \right]$, whose cardinality is $f\left(\tilde{I}\right)=\lfloor \frac{P-\tilde{s}-1}{\tilde{P}}\rfloor +1$, $\lfloor \frac{P}{\tilde{P}}\rfloor -1\leq \lfloor \frac{P-\tilde{s}-1}{\tilde{P}}\rfloor \leq \lfloor \frac{P}{\tilde{P}}\rfloor ,$ and
(14)$$\lfloor \frac{P}{\tilde{P}}\rfloor \leq f\left(\tilde{I}\right)\leq \lfloor \frac{P}{\tilde{P}}\rfloor +1.$$The statement is proven. □

Let us study how the cardinality of the enumeration sets of the Asmuth-Bloom scheme and DT-RRNS are related to RRNS parameters. The Asmuth-Bloom scheme is determined by the set of moduli $p_{0}<p_{1}<p_{2}<\ldots <p_{k}<p_{k+1}<\ldots <p_{n}$. To ensure the asymptotic ideality of the Asmuth-Bloom SSS, we require that this sequence be compact with the initial value $p_{0}$, or $p_{n}<p_{0}+p_{0}^{\theta }$ for θ∈(0,1). In this case, the sequence $p_{1}<\ldots <p_{n}$ will be compact with the initial value $p_{1}$. This RRNS will be used as the basis for the proposed DT-RRNS.

The cardinality of the Asmuth-Bloom enumeration set is constant and equal to $p_{0}$.
(15)$$\lfloor \frac{P}{{\tilde{P}}_{max}}\rfloor \leq f\left({\tilde{I}}_{max}\right)\leq \lfloor \frac{P}{{\tilde{P}}_{max}}\rfloor +1<p_{0}.$$

${\tilde{I}}_{max}$ is a set of unauthorized subsets numbers with the largest range ${\tilde{P}}_{max}$, then,
(16)$${\tilde{P}}_{max}=\overset{k-2}{\underset{i=0}{\prod }}p_{n-i}.$$

Establish a relationship between the value $f\left({\tilde{I}}_{max}\right)$ and k. Let us consider two SSSs with a threshold k and n−k+1, assuming $2\leq k\leq \frac{n}{2}$.

Let us calculate $\frac{P}{{\tilde{P}}_{max}}$ for the second SSS:(17)$$\frac{\prod_{i=1}^{n-k+1}p_{i}}{\prod_{i=0}^{n-k-1}p_{n-i}}=\frac{p_{1}\cdot p_{2}\cdot ...\cdot p_{k}\cdot p_{k+1}\cdot ...\cdot p_{n-k+1}}{.p_{k+1}\cdot p_{k+2}\cdot ...\cdot p_{n-k+1}\cdot p_{n-k+2}\cdot ..\cdot p_{n}}=\frac{p_{1}\cdot p_{2}\cdot ...\cdot p_{k}}{p_{n-k+2}\cdot ...p_{n}}=\frac{\prod_{i=1}^{k}p_{i}}{\prod_{i=0}^{k-2}p_{n-i}}.$$

The expression on the right is the value $\frac{P}{{\tilde{P}}_{max}}$ for the first SSS. Therefore, the values $\frac{P}{{\tilde{P}}_{max}}$ symmetrical in k regarding the meaning $\left\lfloor \frac{n}{2}\right\rfloor$. Let now $2\leq k<\lfloor \frac{n}{2}\rfloor$ and $k_{1}=k+1$. Let now $f_{k_{1}}=\frac{P}{{\tilde{P}}_{max}}$ for a threshold $k_{1}$. Let us estimate the value $f_{k_{1}}$:(18)$$f_{k_{1}}=\frac{\prod_{i=1}^{k_{1}}p_{i}}{\prod_{i=0}^{k_{1}-2}p_{n-i}}=\frac{\prod_{i=1}^{k+1}p_{i}}{\prod_{i=0}^{k-1}p_{n-i}}=\frac{p_{n-k+2}}{p_{k+1}}\cdot \frac{\prod_{i=1}^{k}p_{i}}{\prod_{i=0}^{k-2}p_{n-i}}=\frac{p_{n-k+2}}{p_{k+1}}\cdot f_{k}.$$
where $f_{k}$ represents the value $\frac{P}{{\tilde{P}}_{max}}\;$ for SSS with threshold k. Because $2\leq k<\frac{n}{2}$, then, $n-k+2>\frac{n}{2}$, therefore, n−k+2>k+1. Considering the restrictions imposed on the RRNS moduli, we have $p_{n-k+2}>p_{k+1}$, therefore, $\frac{p_{n-k+2}}{p_{k+1}}>1$.

Given the above considerations, we obtain that $f_{k_{1}}<f_{k}$. In other words, the worst case in which $\frac{P}{{\tilde{P}}_{max}}$ takes the smallest value is the case $k=\lfloor \frac{n}{2}\rfloor$. Because of the symmetry $\frac{P}{{\tilde{P}}_{max}}\;$ relative to this value, it is advisable to consider k within the borders $\left[2,\frac{n}{2}\right]$, as for the interval $\left[\frac{n}{2},n-1\right]$ reasoning proceeds in a similar way. A special case is SSS in which k=n and $\frac{P}{{\tilde{P}}_{max}}=1$.

Next, we prove several important statements that accurately estimate $f\left({\tilde{I}}_{max}\right)$.

**Statement** **3.***For any sequence* $p_{1}<p_{2}<\ldots <p_{n}$ *follows that*(19)$$f\left({\tilde{I}}_{max}\right)=\{\begin{array}{c} \frac{P}{{\tilde{P}}_{max}}+1\;\mathrm{if}\;\tilde{s}<{\left|P\right|}_{{\tilde{P}}_{max}}\;,\\ \frac{P}{{\tilde{P}}_{max}},\;\;\;\;\;\;\;\;\;\;\;\;\;otherwise \end{array}$$

**Proof.** From Statement 2, it follows
(20)$$f\left({\tilde{I}}_{max}\right)=\frac{P-\tilde{s}-1}{\tilde{P}}+1.$$If $\tilde{s}<{\left|P\right|}_{{\tilde{P}}_{max}}$ then $\frac{P-\tilde{s}-1}{\tilde{P}}\geq \frac{P-{\left|P\right|}_{{\tilde{P}}_{max}}}{{\tilde{P}}_{max}}=\lfloor \frac{P}{{\tilde{P}}_{max}}\rfloor$ else $\frac{P-\tilde{s}-1}{\tilde{P}}<\frac{P-{\left|P\right|}_{{\tilde{P}}_{max}}}{{\tilde{P}}_{max}}=\lfloor \frac{P}{{\tilde{P}}_{max}}\rfloor$. Hence if $\tilde{s}<{\left|P\right|}_{{\tilde{P}}_{max}}$ then $\left\lfloor \frac{P-\tilde{s}-1}{\tilde{P}}\rfloor =\lfloor \frac{P}{{\tilde{P}}_{max}}\right\rfloor$ else $\lfloor \frac{P-\tilde{s}-1}{\tilde{P}}\rfloor =\lfloor \frac{P}{{\tilde{P}}_{max}}\rfloor -1$.The statement is proven. □

Expression (20) shows the upper bound for $f({\tilde{I}}_{max}$). To estimate the lower bound $f\left({\tilde{I}}_{max}\right)$, we prove the following statement.

**Statement** **4.***For any sequence* $p_{1}<p_{2}<\ldots <p_{n}$ *such that* $p_{n}<p_{1}+p_{1}^{\theta }$, 0<θ<1, *and any*  k *such that* $2\leq k<\lfloor \frac{n}{2}\rfloor$, *the following inequality is satisfied*
(21)$$f\left({\tilde{I}}_{max}\right)=\frac{P}{{\tilde{P}}_{max}}>p_{1}{\left(\frac{1}{1+p_{1}^{\theta -1}}\right)}^{k-1}.$$

**Proof.** Since the sequence is compact with the initial value $p_{1}$, then ${\tilde{P}}_{max}<{\left(p_{1}+p_{1}^{\theta }\right)}^{k-1}$. On the other hand, since the sequence is increasing, then $P>p_{1}^{k}$. Consequently
(22)$$f\left({\tilde{I}}_{max}\right)=\frac{P}{{\tilde{P}}_{max}}>\frac{p_{1}^{k}}{{\left(p_{1}+p_{1}^{\theta }\right)}^{k-1}}=p_{1}{\left(\frac{p_{1}}{p_{1}+p_{1}^{\theta }}\right)}^{k-1}=p_{1}{\left(\frac{1}{1+p_{1}^{\theta -1}}\right)}^{k-1},$$
from which the inequality (21) follows. The statement is proven. □

Based on Statements 3 and 4, let us accurately determine the boundaries for the quantity $\frac{P}{{\tilde{P}}_{max}}$, which directly depends on the value $p_{1}$.

Let us consider an example that shows how fast the value $\frac{P}{{\tilde{P}}_{max}}$ converges to $p_{1}$.

Figure 4 shows $R=\frac{f\left({\tilde{I}}_{max}\right)}{p_{1}}$. R shows the relation of $f\left({\tilde{I}}_{max}\right)$ and $p_{1}$. This relation assesses how parameter a affects security by approaching the value of b by using compact sequences for various $p_{1}$. We see that with increasing $p_{1}$, R approaches 1, and, therefore, $\frac{P}{{\tilde{P}}_{max}}$ approaches $p_{1}$. In this case, Equation (22) estimates the lower bound of R.

Statement 4 estimates the proximity of the cardinality of possible secrets $f\left({\tilde{I}}_{max}\;\right)$ to $p_{1}$ depending on the $p_{1}$, θ, k, and n given before generating the sequence itself.

Figure 4 shows that the higher the value of $p_{1}$, the closer $f\left({\tilde{I}}_{max}\right)$ to it. Thus, with a higher value of $p_{1}$, the DT-RRNS has higher security.

Statement 4 is important for estimating security. At fixed θ and k, magnitude $f\left({\tilde{I}}_{max}\right)=\frac{P}{{\tilde{P}}_{max}}$ is within the following limits:(23)$$p_{1}{\left(\frac{1}{1+p_{1}^{\theta -1}}\right)}^{k-1}<\frac{P}{{\tilde{P}}_{max}{}_{}}$$

It is easy to show that for fixed 0<θ<1 and 2≤k≤n
(24)$$\underset{p_{1}\to \infty }{\lim}{\left(\frac{1}{1+p_{1}^{\theta -1}}\right)}^{k-1}=1\;\mathrm{and}\;\frac{1}{1+p_{1}^{\theta -1}}<1.$$

Consequently,
(25)$${\left(\frac{1}{1+p_{1}^{\theta -1}}\right)}^{k-1}=\left(1-\varepsilon \right),$$
where 0<ε<1. And the more $p_{1}$, the closer ε to 0. Then from (24) it follows
(26)$$p_{1}\left(1-\varepsilon \right)<\frac{P}{{\tilde{P}}_{max}{}_{}},$$

Based on this expression, one can obtain the following estimate for $f\left({\tilde{I}}_{max}\right)$:(27)$$p_{1}-\varepsilon p_{1}<f\left({\tilde{I}}_{max}\right).$$

The last inequality determines the degree of closeness of the quantity $f\left({\tilde{I}}_{max}\right)$ to $p_{1}$ without generating the sequence itself. Because $p_{1}>p_{0}$ then due to restrictions imposed on $\frac{P}{{\tilde{P}}_{max}}$, with an increasing number $p_{0}$, the cardinality $f\left({\tilde{I}}_{max}\right)$ of the enumeration set of DT-RRNS approaches $p_{1}$. We can conclude that the cardinality of the brute force set for the DT-RRNS when choosing sufficiently large moduli is equivalent to the cardinality of the brute force set of the Asmuth-Bloom scheme, which is equal to $p_{0}$.

Let us now compare the DT-RRNS with the Mignotte scheme. The basic design requirement of the Mignotte scheme is the inclusion of a secret s into the interval $\left(\alpha =\overset{k-2}{\underset{i=0}{\prod }}p_{n-i},\beta =\overset{k}{\underset{i=1}{\prod }}p_{i}\right)$. The statements proved earlier regarding the size of the set of enumerations of the DT-RRNS allow us to deviate from this rule in favor of increasing the dynamic range of the secret representation. Based on the assumption of a uniform distribution of the secret in the interval [0,P), compactness of the set $p_{0}<p_{1}<\ldots <p_{n}$ and a sufficiently large number $p_{0}$, it is easy to show that the probability of a secret falling into the interval [0,α) approaches the probability of “guessing” an arbitrary secret in the Asmuth-Bloom scheme.

Note that in the notation used, $\alpha ={\tilde{P}}_{max}$ and β=P. Indeed, the secret in the Asmuth-Bloom scheme is in the range $\left[0,p_{0}\right)$ and is determined by $p_{0}$. With a uniform distribution of the secret on this set, the probability of choosing an arbitrary secret is $\frac{1}{p_{0}}$. On the other hand, the probability of a number falling into the interval $\left[0,\alpha \right)=\left[0,{\tilde{P}}_{max}\right)$ is equal to $\frac{\left|\left[0,{\tilde{P}}_{max}\right)\right|}{\left|\left[0,P\right)\right|\frac{{\tilde{P}}_{max}}{P}}$. According to Statement 3, for a sufficiently large $p_{0}$, the magnitude $\frac{P}{{\tilde{P}}_{max}}$ is equivalent to $p_{0}$.

It follows that parameters that are determined by DT-RRNS can eliminate restrictions imposed on the parameters of the Mignotte scheme. Let us consider examples of generating DT-RRNS parameters.

**Example** **1.***Let* $p_{1}=1024$, n=10, *and* k=5, *and let it be required that the deviation from the Asmuth-Bloom search power does not exceed 10%. Determine what should be* θ *in this case. According to estimates (24) and (28), we obtain*:(28)$$\theta <\log_{p_{1}}\left(\frac{1}{\sqrt[{k-1}]{1-\varepsilon }}-1\right)+1,$$*where*ε*is a required deviation. In our case,*ε=0.05.

*Substituting the available data into the formula, we have* θ<0.477.


*Consequently, the numbers that provide the required cardinality of the enumeration set must be within the interval*

[1024,1051)

*. Using Statement 4, we have*

$$f\left({\tilde{I}}_{max}\right)>p_{1}{\left(\frac{1}{1+p_{1}^{\theta -1}}\right)}^{k-1}=1024{\left(\frac{1}{1+1024^{0.477-1}}\right)}^{4}\approx 921.762.$$



Fewer unique divisors of $p_{1}$, the more beneficial to use them for building a compact sequence. It increases the number of the coprime numbers in the interval from $p_{1}$ to $2p_{1}$.

It is worth noting that the proof of the possibility of generating compact sequences is a difficult number-theoretic problem.

The generation of a variety of compact sequences is the subject of further research. Now, we can limit ourselves to practical recommendations, which consist in choosing sufficiently large $p_{1}$ and with the least number of divisors.

Statements 1–4 evaluate the security of the DT-RRNS scheme. First, according to Statement 1, $p_{1}>2^{k-1}$ determines a lower bound of $p_{1}$. For maximum security, $p_{1}$ must be significantly higher than $2^{k-1}$. Secondly, an important parameter of the scheme is the value θ, defining a compact sequence. The closer θ to zero, the better the SSS properties in terms of security, which follows from Statement 4 and inequality (24).

## 7. Security of Data Transmission

The RRNS allows the implementation of the integrity, availability, and confidentiality of data by a single mechanism. These features provide an efficient way to ensure reliability and security during data transmission in MANET.

This section discusses the principles on which the proposed method of data transmission in a non-hierarchical network is based.

To meet MANET requirements, we choose a symmetric encryption scheme, a secure RRNS with a compact set of moduli $\left\{p_{1},p_{2},\ldots ,p_{k},p_{k+1},\;\ldots ,p_{n}\right\}$, for which $p_{n}<p_{1}+p_{1}^{\theta }$, where 0<θ<1. To provide the required level of security, the moduli must be close in size to each other.

The combined use of multipath routing, a secure sharing scheme, and the error correction capabilities of RRNS create the conditions for using a new approach to data transport that guarantees transmission reliability and security.

The main principles of the proposed approach:Data are encrypted by a symmetric encryption algorithm and key K.The encrypted data are represented as a set of n RRNS shares by dividing it on moduli $\left\{p_{1},p_{2},\ldots p_{k},\;p_{k+1},\ldots ,p_{n}\right\}$.Key K is divided based on the perfect Asmuth-Bloom scheme to guarantee a high level of key security.Shares of the secret, which consist of a share of the key and data shares, are sent by a separate route that is associated with this modulo and obtained according to an algorithm with the possibility of multipath routing with division by nodes.If some of them could not be delivered within the given waiting period, the receiving node carries out a verification procedure, which is based on the ability of the RRNS to correct and control data integrity.After checking secret shares for correctness and integrity, the receiving node performs a recovery procedure.To recover the original data, the receiver needs to recover the secret key from key shares and decrypt the data using the obtained key.

Figure 5 shows a generalized scheme of the proposed method of data transmission based on encryption, encoding, and data sharing using RRNS. The key is generated first since its size affects the redundancy of the scheme and, therefore, the overall network load.

Shares are moved along one of the previously constructed routes without crossing the nodes. After receiving all or part of the shares of the secret, the receiver recovers the secret by performing the error-correcting decoding procedure. The original secret is obtained by decrypting the data decoded from the RRNS using the encryption key.

To balance the network load, a weighted SSS is used [39]. The route weight, route length, and route reliability (if a secure routing algorithm is used) can be adapted by changing RRNS parameters.

For example, the shortest route can be associated with the largest RRNS modulo. In this case, the message of this route will be the largest, but the transmission along it will be faster. By associating moduli with routes, we can achieve an increase in the quality and speed of transmission and an overall offload of the data transmission network.

The share of the secret that is represented by the smallest modulo carries less information about the original secret relative to the information by the larger modulo. This feature is applicable to change the flow of information to increase the security of data transmission, transporting the smallest share of the secret along the least reliable route according to some criterion.

The proposed approach is characterized by a combination of reliability and security, which are achieved due to several factors. Reliability is based on multipath routing and the RRNS error correction code.

The reliability of a set of routes W depends on the reliability of all constructed routes as follows [3]:(29)$$1-\underset{\omega \in W}{\prod }\left(1-\Pi_{S,D}^{\omega }\left(t\right)\right),$$
where $\Pi_{S,D}^{\omega }\left(t\right)=\underset{\left\{a,b\right\}\in \omega }{\prod }A_{a,b}\left(t\right)$—reliability of a single route ω∈W, which is the product of the availability $A_{a,b}$ of each of the connections between the nodes a and b at a certain point in time t.

We see that with an increase in the number of routes, the reliability of data transmission increases. In addition, RRNS increases the reliability of data transmission due to excessive noise-resistant coding. RRNS controls not only the situation with the loss of availability of an individual node and connectivity but also damage due to failures and intentional distortion of information.

## 8. Security Analysis

Now, let us consider the security of data transmission through MANET by the proposed method. As noted earlier, security is based on the strength of the RRNS-SSS. The computationally secure SSS has a sufficient level of security without leading to high redundancy, unlike ideal SSSs [39]. Due to the properties of RRNS, this scheme allows not only secure data transmission in networks but also load balancing using distributed transmission of data divided into small shares.

The strength of a particular network configuration depends on the resistance of each node to capture, the network topology, the number of node-separated routes built, the configuration of the SSS, and the moduli selection of the RRNS. It is necessary to consider that the condition for data interception (and at the same time confidentiality violation) is the interception of any number of nodes on n or more routes. Because it is not known in advance which nodes will be intercepted, it is impossible to select and exclude a compromised route in the data transmission protocol.

Let us introduce the following notations:

Pr—the probability of secure data transmission when data will not be intercepted during the time interval $T_{0}$.$Pr{\;}_{node}$—the probability of the node attack-resistance (the probability that during the time interval $T_{0}$ the data on the node will not be intercepted).$Q_{z}$—the probability of interception of z nodes.$E_{z}$—the probability of secret loss with z intercepted nodes.$Pr_{z}$—the probability of secure data transmission with z intercepted nodes${\mathrm{node}}_{ij}$,—node j in the route i.Z—the total number of nodes.z—number of intercepted nodes.

Let us consider the probability Pr for the example of the network with the same number of nodes on each route. Note, that it can be extended to the case with an arbitrary number of nodes on each route.

Let us have four possible data transmission routes (Figure 6), each of which has two nodes, with ${\mathrm{Pr}\;}_{\mathrm{node}}=0.99$. We use a suitable RRNS configuration (3,4) with three working and one redundant modulo.

The probability of interception is 1–Pr. The interception of data at any of the nodes of the route means the loss of confidentiality of the data transmitted by this route.

To intercept the secret, at least three different routes must be intercepted (according to the number of moduli k=3, the minimum necessary for recovery). Therefore, the probability of interception when less than three nodes are intercepted is zero.

If exactly three nodes are attacked, then there are two options:

An attacker will be able to recover the original message, for example, if nodes are ${\mathrm{node}}_{11}$, ${\mathrm{node}}_{21}$, ${\mathrm{node}}_{42}$;An attacker will not be able to recover the original message, for example, if nodes are ${\mathrm{node}}_{11}$, ${\mathrm{node}}_{12}$, ${\mathrm{node}}_{22}$.

The number of possible permutations of 8 nodes taken 3 intercepted at a time that leads to loss of secret (we denote this value by $E_{3}$), multiplied by the probability of intercepting exactly three nodes, gives the probability ${\mathrm{Pr}}_{3}$ of intercepting data when intercepting any three nodes:(30)$${\mathrm{Pr}}_{3}=Q_{3}E_{3},$$
where $Q_{3}={\left(1-{\mathrm{Pr}\;}_{\mathrm{node}}\right)}^{3}{\mathrm{Pr}}_{\mathrm{node}}^{5}$—the probability of intercepting exactly three nodes.

In general, the probability $Q_{z}$ of interception z nodes are calculated considering the formula:(31)$$Q_{z}={\left(1-{\mathrm{Pr}\;}_{\mathrm{node}}\right)}^{z}{\mathrm{Pr}}_{\mathrm{node}}^{Z-z}.$$

For example, if $E_{3}=32$, then
$${\mathrm{Pr}}_{3}=32\cdot {\left(1-0.99\right)}^{3}\cdot 0.99^{5}=0.0000304316816$$

If exactly four nodes were intercepted, then there are also two options:

An attacker will be able to recover the original data, for example, if the intercepted nodes are ${\mathrm{node}}_{11}$, ${\mathrm{node}}_{12}$, ${\mathrm{node}}_{21}$, ${\mathrm{node}}_{42}$;The attacker will not be able to recover the original data, for example, if the intercepted nodes are ${\mathrm{node}}_{11}$, ${\mathrm{node}}_{12}$, ${\mathrm{node}}_{21}$, ${\mathrm{node}}_{22}$.

The number of combinations $E_{4}$ of the four captured nodes, allowing you to recover the original data, multiplied by the probability of intercepting exactly four nodes, will give the probability of intercepting data if any four nodes are intercepted:(32)$${\mathrm{Pr}}_{4}=Q_{4}E_{4}$$

For example, if $E_{4}=64$ then ${\mathrm{Pr}}_{4}=6.147814464\times 10^{-7}$.

Special attention deserves the case if five or more nodes are intercepted. In the described situation, any set of captured nodes will provide attackers with a means to recover the original data. For situations of this kind, the number of combinations of received nodes that are needed to recover the original message will be equal to the total number of permutations with repetitions of 8 nodes of 5, 6, 7, and 8, respectively. Calculated values: $E_{5}=56$, $E_{6}=28$, $E_{7}=8$, $E_{8}=1$, Then, guided by the approach proposed earlier, we obtain that $Pr_{5}=5.434\times 10^{-9}$, $Pr_{6}=2.744\times 10^{-11}$, $Pr_{7}=7.92\times 10^{-14}$ и $Pr_{8}=10^{-16}$.

Using the results of probability calculations for each of the cases, it is possible to isolate the overall probability of intercepting data:$$1-Pr=\overset{8}{\underset{z=3}{\sum }}Pr_{z}=0.000031052$$

It turns out that the probability Pr of secure data transmission is Pr=0.999968948 for ${\mathrm{Pr}}_{\mathrm{node}}=0.99$.

Table 1 shows the probability Pr of secure data transmission in MANET with redundant (3,4)-RRNS four possible data transmission routes and two nodes on each of the routes, for different values of $Pr_{node}$.

Table 1 shows that for z=0,1,2 the probability $Q_{z}$ is high. However, data transmitted by DT-RRNS are not intercepted. If z≥3 and ${\mathrm{Pr}}_{\mathrm{node}}\geq 0.7$, probability $Q_{z}\leq 4.537\times 10^{-3}$, which reduces the probability of secure data transmission ${\mathrm{Pr}}_{z}$.

We note that with increasing the number of possible routes and corresponding changing RRNS parameters, the probability of a secure data transmission increases.

Table 2 shows that the probability of a secure transmission grows quite fast with increasing (k, n) parameters and number of nodes, even for the high probability of the resistance of the node to data interception ${\mathrm{Pr}}_{\mathrm{node}}=0.99$.

## 9. Performance Analysis

In this section, we compare two SSS-RRNS solutions: well-known Asmuth-Bloom and DT-RRNS. To measure encoding time, decoding time, and redundancy, we transmit data from 6 MB to 146 MB across a network of 16–24 nodes with 4 neighboring nodes, using a number of moduli from 4 to 6. The secret key used for the schemes is 2147483659.

Table 3 and Figure 7 show the encoding and decoding time, and redundancy for the Asmuth-Bloom and DT-RRNS schemes with varying data sizes and the number of moduli. We see that the encoding and decoding times increase linearly for both Asmuth-Bloom and DT-RRNS.

DT-RRNS shows better runtime results than Asmuth-Bloom. DT-RNS for the largest data size has less time than Asmuth-Bloom for the smallest data size when using 6 moduli. The redundancy for DT-RRNS is approximately the same for all moduli sets and data sizes. The redundancy of the Asmuth-Bloom is increasing with the number of moduli increasing.

Table 4 contains the moduli used in the experiment.

Figure 7 shows the encoding time (a) and decoding time (b) versus the number of moduli and data size. Figure 8 shows the redundancy versus the number of moduli and data size. DT-RRNS has lower redundancy close to 1. We see that the redundancy of both Asmuth-Bloom and DT-RRNS weakly depends on the input data. It varies with scheme parameters.

We see that the proposed DT-RRNS has several advantages. It increases the speed of the system. The encoding time is in the range of 1 to 180 ms, while Asmuth-Bloom is between 40 and 1100 ms. It has reduced data redundancy while maintaining the same level of security and reliability.

## 10. Concluding Discussion

Large-scale data sharing in a distributed smart city environment requires an increased attention to data security and reliability issues. Methods that ensure data security at the required level with high reliability and speed are very important.

In this work, we propose a DT-RRNS routing solution for the WSN and MANET complex dynamic hierarchical heterogeneous networks for improving data transmission. To design efficient mechanisms, we consider reliability and security as close concepts. Increased security and reliability are achieved with an effective data recovery mechanism of RRNS with moduli of compact sequences of coprime numbers.

This mechanism together with adaptive multipath routing increases the resistance of the sensor network to attacks of various types, including unauthorized interception, message falsification, errors, node and network connection failures, information loss in case of attacks or accidents, etc.

This approach does not have the limitations of the traditional encryption methods for secured data transmission. The secret key management is solved by the SSS.

In addition, this solution reduces data redundancy, resulting in less use of large equipment, energy consumption, and message storage capacity. These properties are important when deploying IoT.

In the DT-RRNS, each participant receives shares of a smaller size than the original data. It improves transmission speed, resulting in better support for big data sensing and processing, in contrast to the Asmuth-Bloom scheme.

The promising direction for future work is the development of computationally efficient methods for generating dynamic RRNS parameters and dynamic routes due to loss of sensors, connections, loss of functionality, errors by contamination, vibration, shocks, high temperatures, etc. It is important to study the problem of selecting moduli for dynamic adaptation to changing network topology and characteristics. To further improve efficiency and reliability, we will consider specialized multipath routing protocols based on a weighted version of DT-RRNS.

## Figures and Tables

**Figure 1 sensors-23-03738-f001:**
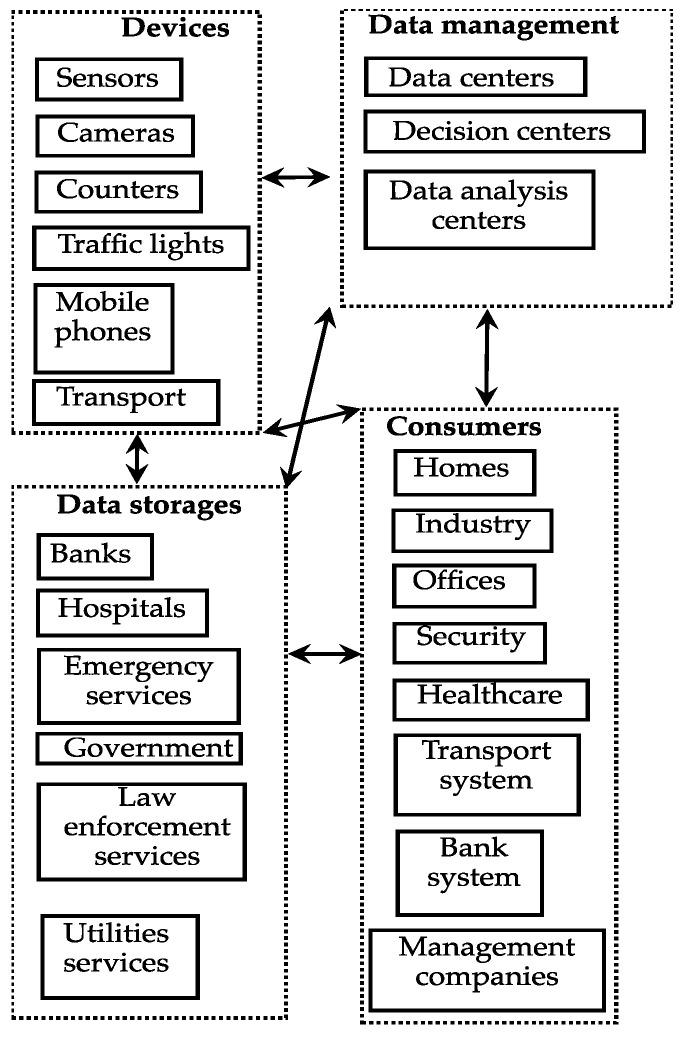
The general model of data transmission.

**Figure 2 sensors-23-03738-f002:**
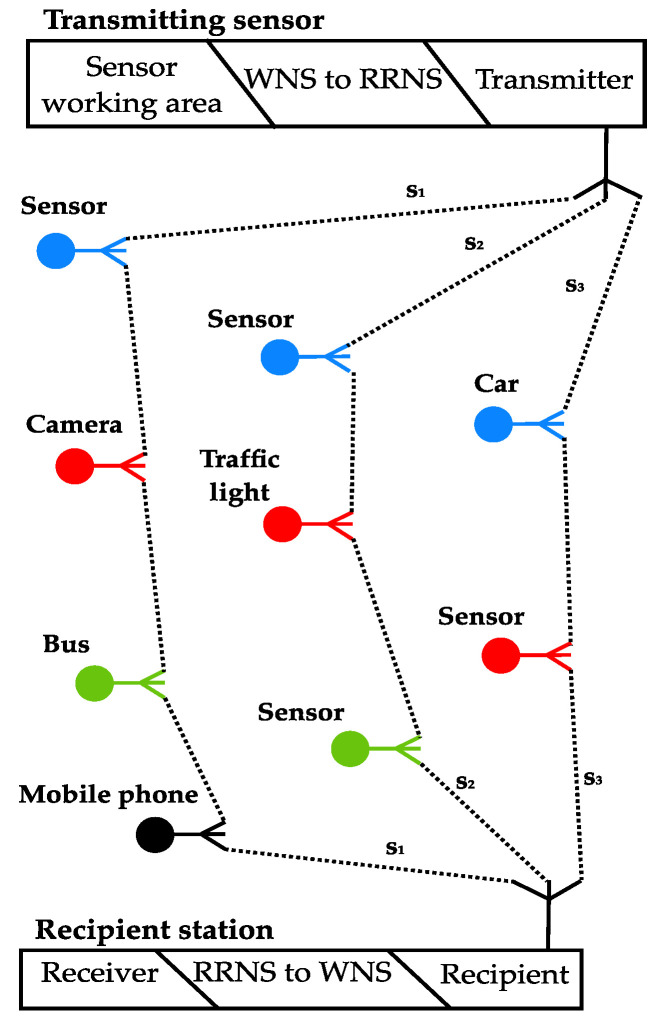
Distributed data transmission model.

**Figure 3 sensors-23-03738-f003:**
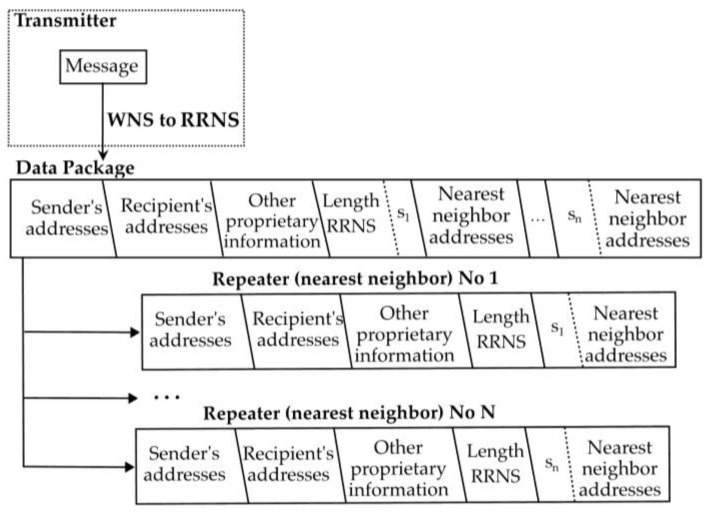
Data packet model.

**Figure 4 sensors-23-03738-f004:**
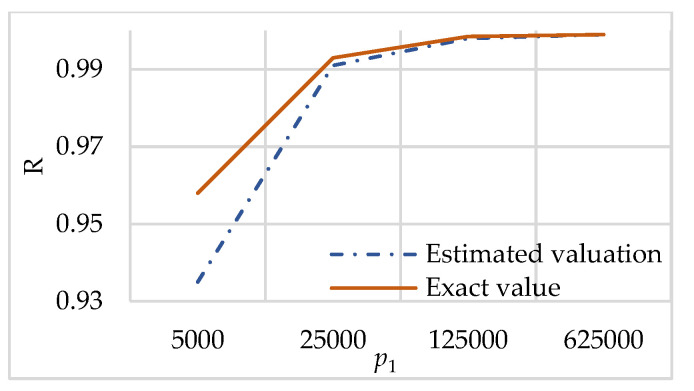
$R=\frac{f\left({\tilde{I}}_{max}\right)}{p_{1}}$ varying $p_{1}$ for θ=0.477, n=15, and k=7.

**Figure 5 sensors-23-03738-f005:**
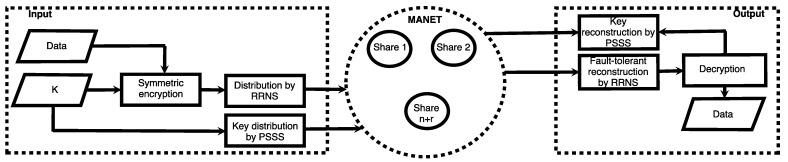
A generalized scheme for secure and reliable data transmission based on a computationally secure SSS.

**Figure 6 sensors-23-03738-f006:**
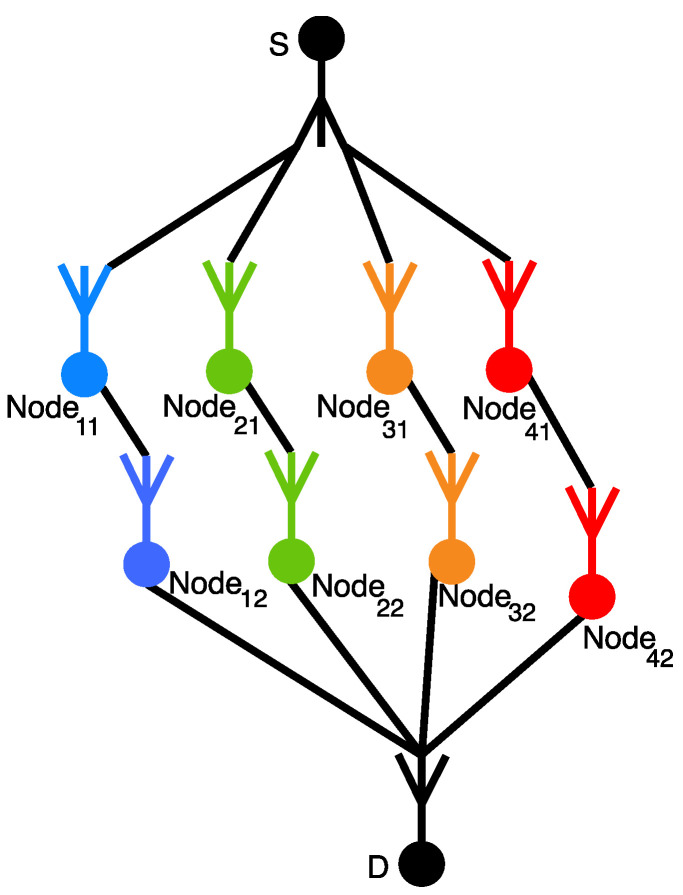
Routing example for 4 тon-crossing routes with 2 nodes per each route.

**Figure 7 sensors-23-03738-f007:**
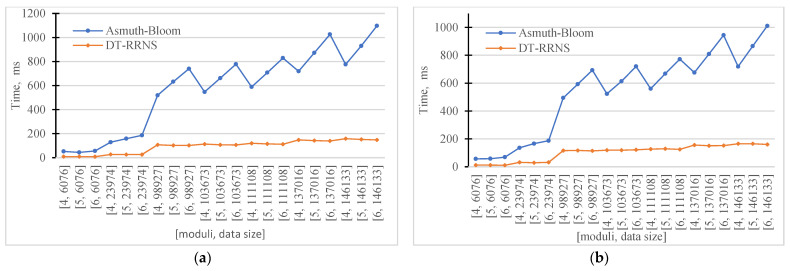
Encoding time and obtained redundancy for Asmuth-Bloom and DT-RRNS (**a**) Encoding time. (**b**) Decoding time.

**Figure 8 sensors-23-03738-f008:**
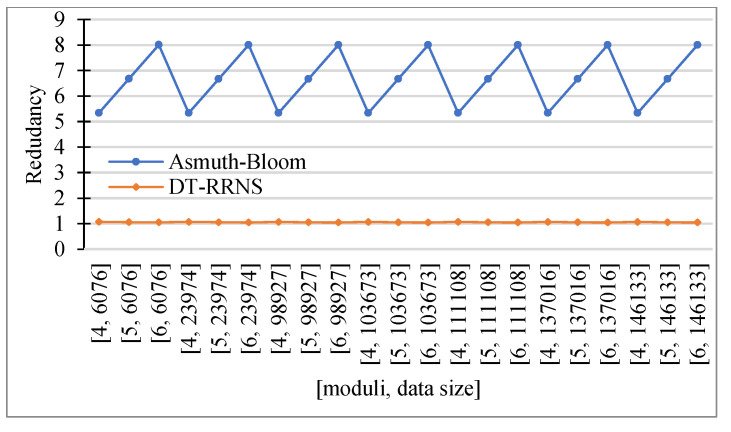
Redundancy.

**Table 1 sensors-23-03738-t001:** Probability of secure data transmission $Pr_{z}$ with z intercepted nodes for the different probability of the node attack resistance.

z	$Pr_{node}=0.7$		$Pr_{node}=0.9$		$Pr_{node}=0.99$	
	$Pr_{z}$	$Q_{z}$	$Pr_{z}$	$Q_{z}$	$Pr_{z}$	$Q_{z}$
0	1	$5.757\times 10^{-2}$	1	$4.304\times 10^{-1}$	1	$9.22\times 10^{-1}$
1	1	$2.47\times 10^{-2}$	1	$4.782\times 10^{-2}$	1	$9.32\times 10^{-3}$
2	1	$1.05\times 10^{-2}$	1	$5.314\times 10^{-3}$	1	$9.4\times 10^{-5}$
3	$1.452\times 10^{-1}$	$4.537\times 10^{-3}$	$1.88\times 10^{-4}$	$5.9\times 10^{-4}$	$3.04\times 10^{-5}$	$9.509\times 10^{-7}$
4	$1.244\times 10^{-1}$	$1.944\times 10^{-3}$	$4.199\times 10^{-4}$	$6.5\times 10^{-5}$	$6.147\times 10^{-7}$	$9.605\times 10^{-9}$
5	$4.667\times 10^{-2}$	$8.33\times 10^{-3}$	$4.08\times 10^{-4}$	$7\times 10^{-6}$	$5.433\times 10^{-9}$	$9.702\times 10^{-11}$
6	$10^{-2}$	$3.57\times 10^{-4}$	$2.2\times 10^{-5}$	$8.1\times 10^{-7}$	$2.744\times 10^{-11}$	$9.801\times 10^{-13}$
7	$1.224\times 10^{-3}$	$1.53\times 10^{-4}$	$7.2\times 10^{-7}$	$9\times 10^{-8}$	$7.92\times 10^{-14}$	$9.9\times 10^{-15}$
8	$6.56\times 10^{-3}$	$6.5\times 10^{-5}$	$10^{-8}$	$10^{-8}$	$10^{-16}$	$10^{-16}$
	$Pr=6.723\times 10^{-1}$		$Pr=9.764\times 10^{-1}$		$Pr=9.999\times 10^{-1}$	

**Table 2 sensors-23-03738-t002:** Probability Pr of secure data transmission at different numbers of routes and the total number of nodes.

(k, n)/Number of Nodes	Pr	
	$Pr_{node}=0.9$	$Pr_{node}=0.99$
(2, 3)/6	0.905	0.998827731
(3, 4)/8	0.976	0.999968948
(4, 5)/10	0.994	0.999999228
(5, 6)/12	0.998	0.999999982
(6, 7)/14	0.9997	0.999999999
(7, 8)/16	0.9999	0.999999999

**Table 3 sensors-23-03738-t003:** The coding and decoding time and redundancy.

Scheme	Moduli	Data Size (KB)	Coding Time (ms)	Redundancy	Decoding Time (ms)
Asmuth-Bloom	4	6076	52	5.33772	57
DT-RRNS			9	1.07176	12
Asmuth-Bloom		23,974	129	5.33445	136
DT-RRNS			27	1.06782	32
Asmuth-Bloom		98,927	519	5.33355	494
DT-RRNS			107	1.06697	116
Asmuth-Bloom		103,673	547	5.33354	523
DT-RRNS			113	1.06689	119
Asmuth-Bloom		111,108	589	5.33348	560
DT-RRNS			120	1.06693	127
Asmuth-Bloom		137,016	719	5.33345	676
DT-RRNS			148	1.06685	156
Asmuth-Bloom		146,133	776	5.33344	719
DT-RRNS			158	1.06686	165
Asmuth-Bloom	5	6076	44	6.67215	58
DT-RRNS			9	1.05662	12
Asmuth-Bloom		23,974	158	6.66806	166
DT-RRNS			26	1.05364	29
Asmuth-Bloom		98,927	632	6.66694	593
DT-RRNS			102	1.0529	117
Asmuth-Bloom		103,673	662	6.66692	614
DT-RRNS			107	1.05293	119
Asmuth-Bloom		111,108	708	6.66685	668
DT-RRNS			115	1.05285	129
Asmuth-Bloom		137,016	872	6.66681	809
DT-RRNS			142	1.05287	151
Asmuth-Bloom		146,133	930	6.6668	866
DT-RRNS			152	1.05288	165
Asmuth-Bloom	6	6076	56	8.00658	69
DT-RRNS			8	1.05069	11
Asmuth-Bloom		23,974	186	8.00167	187
DT-RRNS			26	1.04513	32
Asmuth-Bloom		98,927	740	8.00032	693
DT-RRNS			102	1.04392	114
Asmuth-Bloom		103,673	778	8.00031	720
DT-RRNS			106	1.04382	122
Asmuth-Bloom		111,108	829	8.00022	772
DT-RRNS			112	1.04374	125
Asmuth-Bloom		137,016	1026	8.00018	944
DT-RRNS			139	1.04379	152
Asmuth-Bloom		146,133	1097	8.00016	1011
DT-RRNS			148	1.04371	160

**Table 4 sensors-23-03738-t004:** Number of moduli and their meanings.

Number of Moduli	Moduli
4	(2147483693, 2147483713, 2147483743, 2147483777)
5	(2147483693, 2147483713, 2147483743, 2147483777, 2147483783)
6	(2147483693, 2147483713, 2147483743, 2147483777, 2147483783, 2147483813)

## Data Availability

Not applicable.

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
