# Peer review of "DT-RRNS: Routing Protocol Design for Secure and Reliable Distributed Smart Sensors Communication Systems†"

_sensors, 2023, doi:10.3390/s23073738_

Round 1

Reviewer 1 Report

The authors make a Routing Protocol Design for Secure and Reliable  Distributed Smart Sensors Communication Systems. This design provides an efficient mechanism for a Distributed dynamic heterogeneous network Transmission (DT) with new security and reliability routing protocol (DT-RRNS).and improves the reliability and fault tolerance of the entire system. This work is interesting, and I think it is suitable for Sensors after revision. However, there are still some problems that need to be revised as follows.

1.      Please give the description of RRNS.

2.      In table 2, why didn't you talk about Prnode=0.97.

3.      In section 8,why comparing two implementations of SSS-RRNS: Asmuth-Bloom and DT-RRN,and maybe there is a writing error.

4.     Fig7 and Fig8, Why the number of curves on the right is less than that on the left.

Author Response

The authors make a Routing Protocol Design for Secure and Reliable  Distributed Smart Sensors communication Systems. This design provides an efficient mechanism for a Distributed dynamic heterogeneous network Transmission (DT) with new security and reliability routing protocol (DT-RRNS).and improves the reliability and fault tolerance of the entire system. This work is interesting, and I think it is suitable for Sensors after revision. However, there are still some problems that need to be revised as follows.

  • Thank you for the evaluation of our paper.

Please give the description of RRNS.

  • We provide a more detailed presentation of RRNS and basis to configure parameters to cope with security and reliability.

In table 2, why didn't you talk about Prnode=0.97.

  • - This table shows how the probability of a secure transmission grows with increasing numbers of routes and the total number of nodes for Prnode =0.97 and Prnode =0.99. We clarify it in the text.

In section 8,why comparing two implementations of SSS-RRNS: Asmuth-Bloom and DT-RRNS, and maybe there is a writing error.

  • Thank you for your comment. We clarify in the text that we compare our solution with an asymptotic ideal and perfect Asmuth-Bloom scheme which was proved to guarantee a high level of security.

Fig7 and Fig8, Why the number of curves on the right is less than that on the left.

  • Following your recommendations, we completely redesign Figures 7 and 8 to show in more details how time and redundancy depend on data size and number of moduli.

Reviewer 2 Report

1. The authors have claimed that the proposed method increases the speed because the message is divided into smaller parts! Increasing the number of messages means more overhead for the network. On the other hand, it increases the chance of the message being lost or damaged. On what basis and on which authority is this claim made? It must be described. The increase in the number of packets shows its effect especially in high network traffic when many nodes are sending.

2. Equations must be numbered.

3. It is better to clearly add a time diagram of the messages that must be exchanged between the nodes and their content to determine the paths, recover the packets and decode the packets to the article.

4. If possible, the proposed method should be analyzed in terms of time complexity.

5. The conducted tests are not convincing for the reliability, high security and high speed of the method. The details of the experiments should be described more. In a space with what dimensions, with how many nodes in the network, with what arrangement and with what volume for data transmission, the results have been obtained?

6. Communication in sensor networks is usually wireless and broadcast, so all messages sent to different routes can be eavesdropped by an intermediate node. Therefore, explain how segmenting the message into small parts and sending through different routes helps security?

7. Scenarios to check the security of the proposed method should be designed, and the proposed method should be tested based on them.

8. The proposed distributed algorithm should be fully expressed in the form of a flowchart or pseudocode.

Author Response

Comments and Suggestions for Authors

  • We appreciate your evaluation.

The authors have claimed that the proposed method increases the speed because the message is divided into smaller parts! Increasing the number of messages means more overhead for the network. On the other hand, it increases the chance of the message being lost or damaged. On what basis and on which authority is this claim made? It must be described. The increase in the number of packets shows its effect especially in high network traffic when many nodes are sending.

  • We agree with the reviewer. The data transmission performance in real systems depends on many factors such as network traffic, bandwidth, network congestion, latency, etc. We provide theoretical analysis of the influence of the number of messages on the probability of a packet intercepting, lost, or damaging. We clarity how transmitting speed is increased by reducing the size of the messages together with parallel transmission. We also present the proof of the level of fault tolerance using (k n) threshold secret sharing schemes and we have detailed positive properties of the DT-RRNS.

Equations must be numbered.

  • - done

It is better to clearly add a time diagram of the messages that must be exchanged between the nodes and their content to determine the paths, recover the packets and decode the packets to the article.

  • We agree with the reviewer that it is important to consider time diagrams and complexities of real MANET network functions in different scenarios including path finding algorithms, nodes interactions protocols, overheads, etc. This will be a part of future work. In this paper, we presented a generalized model of a message transmission for studying theoretical concepts of the proposed protocol.

If possible, the proposed method should be analyzed in terms of time complexity.

  • Thank you for your comment, it is an important issue and will be studied in future works.

The conducted tests are not convincing for the reliability, high security and high speed of the method. The details of the experiments should be described more. In a space with what dimensions, with how many nodes in the network, with what arrangement and with what volume for data transmission, the results have been obtained?

  • - Following your recommendations, we explain these issues in the text and describe theoretical estimates and experimental results in details

Communication in sensor networks is usually wireless and broadcast, so all messages sent to different routes can be eavesdropped by an intermediate node. Therefore, explain how segmenting the message into small parts and sending through different routes helps security?

  • We agree and explain in more details that simple message segmentation into small pieces and sending them along different routes does not bring security benefits. RRNS coding, recovery mechanism, with properties of SSS provide security and reliability discussed in Sections 4-8. Even if attackers intercept several secret shares and secret key shares, they cannot overcome the secret recovery threshold and do not receive any useful information about the secret.

Scenarios to check the security of the proposed method should be designed, and the proposed method should be tested based on them.

  • We agree with the reviewer. In the manuscript, we did not discuss security verification in real scenarios. This will be the subject of future work.

The proposed distributed algorithm should be fully expressed in the form of a flowchart or pseudocode.

  • Our work is based on the known algorithms presented in References. They are well studied and discuss flowcharts and pseudocodes. We study reliability and security aspects of these algorithms applied in a special way, as well as the proof of fundamental statements.

Reviewer 3 Report

Authors have proposed a DT-RRNS routing solution for the WSN and MANET complex dynamic hierarchical heterogeneous networks with data transfer acceleration and protection. 

Comments:

1. It is suggested to include the main contributions (as bullet points) of the proposed work in the Introduction section. 

2.  Suggest adding more recent papers (2023) in the Related work Section, if available. 

3. Section 8- Experimental Analysis-  Suggest increasing the clarity of the graphs. Now, it is difficult to understand these graphs. You may use colours. 

4. English checking is recommended. 

Author Response

Authors have proposed a DT-RRNS routing solution for the WSN and MANET complex dynamic hierarchical heterogeneous networks with data transfer acceleration and protection. 

  • Thank you for the evaluation of our paper.

Comments:

It is suggested to include the main contributions (as bullet points) of the proposed work in the Introduction section. 

  • Thank you for the observation, we include discussion of the main contributions in the introduction and summarized in the conclusion.

Suggest adding more recent papers (2023) in the Related work Section, if available. 

  • We analyze recent results related to the research topic and include up-to-date references.

Section 8- Experimental Analysis Suggest increasing the clarity of the graphs. Now, it is difficult to understand these graphs. You may use colors. 

  • Following your recommendations, we reorganize experimental analysis sections to clearly explain reliability, security, and performance results. We completely redesign Figures 7 and 8, to show in more details how time and redundancy depend on data size and number of moduli.

English checking is recommended. 

  • We have cleaned up the paper, and very carefully proofread it to improve English and clarity.